# Firm Sustainable Development Goals and Firm Financial Performance through the Lens of Green Innovation Practices and Reporting: A Proactive Approach

**Parvez Alam Khan [1], Satirenjit Kaur Johl [1,*] and Shakeb Akhtar [2]**

[1] Department of Management and Humanities, Universiti Teknologi PETRONAS, Seri Iskandar 32610, Perak, Malaysia; parvezkhan.alam@gmail.com

[2] Department of Finance, School of Business, Galgotias University, Greater Noida 203201, India; shakebakhtar.amu@gmail.com

* Correspondence: satire@utp.edu.my

**Abstract:** The current global economy demands synergy between ecological responsiveness and proactive business models. To analyze these dynamics, the objective of this study is to simultaneously investigate the effects of green innovation practices concerning the sustainable development goals (SDG) and financial performance of firms. This study also advocates for the injection of green innovation reporting into sustainable reporting for greater disclosure. Data from sixty-seven companies from five continents and the top five blue chip firms for each country are collected through content analysis, with the generalized least squares (GLS) approach used to test a causal relationship hypothesis. The results indicate mixed findings, with green product innovation showing positive relationships with returns on equity (ROE) and returns on investments (ROI). At the same time, green process innovation shows negative relationships with returns on assets (ROA) but shows a positive impact on returns on investments (ROI) and firm SDGs. In contrast, green service innovation shows an insignificant relationship with financial performance and SDGs. On the other hand, non-operational green innovation variables and green marketing positively affect returns on assets and investment, showing significant negative impacts on returns on equity. However, green organizational innovation shows an insignificant relationship with firm financial performance and SDGs. In addition, this study also shows that the Australia/New Zealand region is the leader in green innovation reporting, followed by Europe, Asia, Africa, and lastly, North America.

**Keywords:** green innovation; sustainable development goal; environmental policy; financial performance; green innovation disclosure

## 1. Introduction

In the last three decades, responsible and sustainable investment has been seen as an investment revolution across various economies. The nature of responsible investment has undergone a massive shift due to the integration of environmental, social, and governance factors (ESG) into investment strategies (Duric and Topler 2021; Gifford 2016). The integration of environmental, social, and governance factors into socially responsible investment (SRI) has brought significant market returns to investors (Leins 2020; Vo et al. 2019).

Equally, the introduction of 17 sustainable development goals (SDGs) by the United Nations has created new prospects for socially responsible investors around the globe (García-Sánchez et al. 2020; Lu et al. 2021). With the incorporation of ESG and SDGs into responsible investments, investor activity has increased, which can be traced to the Japanese Sustainable Investment Forum report, where Japanese SRI jumped from 3.4% to 242% in 2017, along with sustainable and responsible impact investing in the U.S., which rose to USD 12.0 trillion. Globally, responsible investment has increased to 34% over the past two years.

However, there are still severe concerns regarding environmental and societal challenges due to increases in greenhouse gas rates, waste generation, wastewater, climate change, and biodiversity loss. These increases in global warming, natural resource depletion, ocean acidification, ozone layer depletion, and the use of limited resources will severely affect the fulfillment of resource demands in the next generation (World Bank 2018).

These global concerns for the environment within the context of the current industrial revolution 4.0 (Alaloul et al. 2020) have drawn the attention of various industrialists, academics, governments, and other related institutions, who are challenging companies to embrace lean culture or green culture innovations (Duarte and Cruz-Machado 2013). In reference to existing industry practices, lean and green culture innovations represent a modern approach to minimizing current environmental issues. If this lean and green culture is injected into innovation, then the current innovation practices will help companies save time and money by accelerating resource conservation, establishing more sustainable systems, gaining a strategic edge, and generating extra revenue. Hernandez-Vivanco et al. (2018) suggested that green innovation strategies are important for firms in minimizing their ecological effects and creating sustainable goodwill.

In addition, an extensive literature review was conducted to highlight research gaps in this field, which showed that various studies have explored the relationships between green innovation, firm performance (Zhang et al. 2020; Palčič and Prester 2020; Khan and Johl 2019), and other organizational support aspects such as human resources (Singh et al. 2020), leadership style (Zhou et al. 2018), knowledge acquisition (Wang et al. 2020), technology adoption, ISO certification (Khan and Johl 2019), and innovation culture (Yang et al. 2017) with mixed findings. The literature search showed a lack of research on green innovation (Khan et al. 2021). Additionally, limited studies have been conducted on green innovation and reporting together with firm sustainable development goals and financial performance.

Therefore, the main objective of this study was to explore and expand on the research by assessing the knowledge gaps regarding green innovation practices and green innovation reporting (expanding on the study of green innovation reporting conducted by Khan et al. 2021a) of operational and non-operational activities related to the performance of financial and sustainable development goals (economic and environmental goal) across five continents.

This study defines green innovation as a combination of green operational innovation and green non-operational innovation. Green operational innovation consolidates green product innovation, green process innovation, and green service innovation, enhancing efficiency and the effective use of available resources and reducing operating costs. Green non-operational innovation includes organizational innovation and marketing innovation, saving resources during organizational activity, further minimizing a business operation's supporting activity costs.

This study contributes to the literature in four ways. Firstly, green innovation has been positively inclined toward firm performance in the short term. Green innovation is observed to have a significant positive impact on a firm's sustainable development goal performance, positively contributing towards achieving the United Nations' Sustainable Development Goals for societal development. Secondly, this study also highlights that the new element of green innovation reporting (disclosure) significantly affects firm and SDG performance and enhances disclosure transparency, which helps attract and retain sustainable investors.

Thirdly, this study provides practitioners and policymakers with insights into how to include green innovation reporting as a new variable in integrated or sustainable reporting, which leads to greater transparency and accountability to the policymakers. This will help policymakers understand the true background for formulating new policies.

Lastly, this research signals the Global Reporting Initiative (GRI) (a reporting standard followed by various countries) that green innovation is missing in their latest standard of

GRI reporting. Global reporting initiatives can also develop another standard for reporting innovation or green innovation to enhance a firm's greater disclosure and transparency.

The article is divided into six sections: Section 2 highlights the literature on each variable, followed by data methodology in Section 3. Section 4 discusses preliminary results, and Section 5 focuses on green innovation disclosure based on five continents. The last section comprises the discussion and the implications of the study.

## 2. Literature and Conceptual Framework Development

Currently, the literature on innovation has reported different types of innovation, which can be categorized as environmentally friendly innovation (green innovation, design-driven innovation, responsible innovation) (Putri and Soewarno 2020), collaboration innovation (open innovation) (Leckel et al. 2020), social innovation (Oeij et al. 2019; Pel et al. 2020), user innovation (Bradonjic et al. 2019), disruptive innovation (Curry et al. 2021), common innovation, convergence innovation (Lee 2015), indigenous innovation (Chen et al. 2020), total innovation (Al-Moaz and Shahein 2019), secondary innovation and embracing innovation (Chen et al. 2018). However, the above types of innovation can be categorized into two types—operational innovation and non-operational innovation—which support the operation's innovation activities and enhance soft and hard qualities (Ali and Johl 2021) without compromising the firm's performance.

### 2.1. Sustainable Development Goals and the Firm's Financial Performance

The resource-based theory is popularly accepted among management scholars, advocating the effective and efficient utilization of organizational resources. The proper usage of limited resources minimizes costs and boosts the firm's performance. Several studies (Tang et al. 2018; Zhang et al. 2019) on green innovation and firm performance have been conducted, showing a positive correlation between the two (Yi et al. 2021).

The performance measurement parameter will be limited to the accounting ratio and environmental performance of the businesses. This study focuses on financial performance and performance with regard to the United Nations' sustainable development goals (SDG) as measurement parameters. The sustainable development goals include environmental goals, social goals, and economic goals (Jan et al. 2021) and companies are adopting SDGs to create a competitive advantage over their competitors. The sustainability of the organization can be assessed through the organization's contribution to the United Nations' 17 SDGs.

The contribution to the United Nations' 17 SDGs can be measured through the company's reporting, initiatives, and investments. The United Nations has set 17 agendas for sustainability for the world to be achieved by 2030. In the 17 SDGs, there are some economic goals (G17 and G9), social goals (G16, G15, G11, G10, G8, G5, G4, G3, G2, G1), and environmental goals (G6, G7, G12, G13, and G14). In this study, green innovation is measured through economic and ecological goals stated in Figure 1.

Firm Financial Performance

Firm financial performance is measured through return on investment (ROI), return on assets (ROA) (Akhtar et al. 2020), and return on equity (ROE) (Yi et al. 2021). All three are accounting ratios that measure firm management efficiency concerning shareholder equity, firm investment, and asset utilization. Furthermore, the three ratios are the most adopted in the literature on sustainability, green innovation (Xu et al. 2021; Yi et al. 2021), and eco-innovation (Johl and Toha 2021; González-Ruiz et al. 2018; Sharif and Alhiyasat 2018), and include sustainable practices (Jan et al. 2021a, 2021b) and the inclusion of intellectual capital into the green board (Shah et al. 2021). Therefore, this study has adopted all three accounting ratios and the new non-financial parameter, i.e., sustainable development goals.

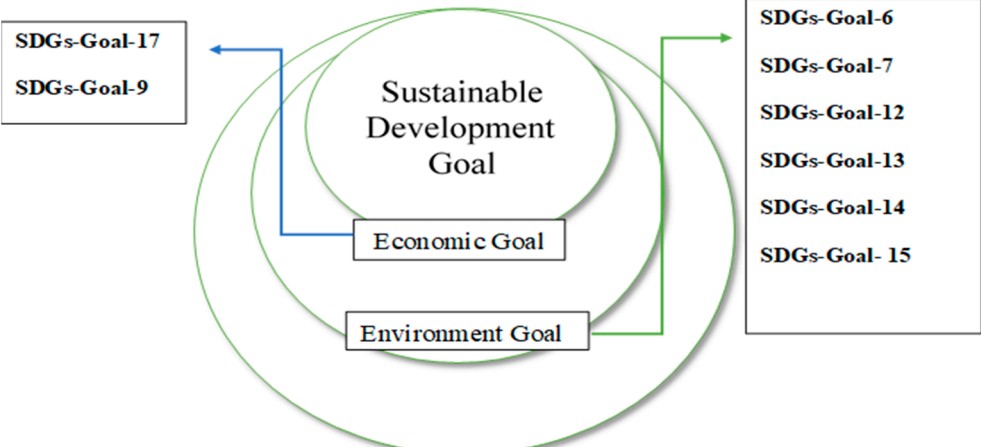

**Figure 1.** Sustainable Development Goal—Economic and Environmental Goal.

### 2.2. Green Operational Innovation (GOI), SDGs, and Financial Performance

Green operational innovation is the modification and introduction of a different product, process, and service that further minimizes emissions and inputs, leading to a green ecology (Calza et al. 2017). Green operational innovation is defined as an innovation that enhances the organization's efficiency of business operation activity. The business operational efficiency can be achieved through green innovation in the business's products, processes, and services. The current literature on green innovation has different synonyms such as ecological, environmental, and eco-innovation, which are used by various researchers (Xie et al. 2019) to integrate with the performance measurement criteria of different contexts. Ultimately, most of the research findings suggest that green innovation decreases the environmental burden, creates value for related stakeholders, and strengthens business activities.

The literature on green innovation has evolved in the current industrial revolution due to the mounting environmental threats in the third and fourth industrial revolutions (Yin et al. 2018; Chen 2008). Scholars Albort-Morant et al. (2016) explored how green innovation practices provide businesses with a way to gain a competitive edge while also providing multiple benefits such as green goodwill, stakeholder confidence, and a high price on the share, which is especially true if the firm is the first mover (Chen 2008; Jabbour et al. 2016; Klingenberg et al. 2013; Calabrese et al. 2018; Tariq et al. 2017; Albort-Morant et al. 2016). Furthermore, Chen (2008) added that the firm's effective and efficient performance and profitability could be achieved. Thus, it benefits the company in two ways: ecologically and financially.

In the existing literature, the various forms of green operational innovation discussed are green product innovation (Tariq et al. 2017; Calza et al. 2017; Xie et al. 2019), green process innovation (Khan et al. 2021), and green service innovation (Calabrese et al. 2018; Khan et al. 2021; Xie et al. 2019; Tariq et al. 2019; Asni and Agustia 2021; Khan and Johl 2020; Calabrese et al. 2018). The main variable of green operational innovation has been investigated over a firm's operational (Jabbour et al. 2016) and financial performance (Klingenberg et al. 2013), particularly regarding returns on investment (Khan and Johl 2020; Tang et al. 2018), returns on assets, and returns on equity (Asni and Agustia 2021; Hu et al. 2021; Qiu et al. 2020; Tariq et al. 2019). However, there is limited research that has investigated the relationship between green innovation and returns on investment, and returns on assets and returns on equity, as these accounting ratios measure the stakeholder perspective (ROE), firm book value (ROI and ROA), and firm management efficiency (ROI, ROE, ROA). This study includes the utmost aspects of green innovation that are related to the operational activity of business found in the literature.

2.2.1. Green Product Innovation (GPI)

GPI has bestowed numerous benefits among stakeholders during the innovation lifecycle. However, the development of green products is slow to meet future expectations, as (Ilg 2019) stated. Green product innovation emboldens the economic and productive use of defined resources, and it depreciates waste to generate added earnings and fund flows (Khan et al. 2021; Rehman et al. 2021).

Green product innovation also creates green goodwill, builds a unique market position, increases competitive advantage, and builds a green leadership reputation. It has become an immense benefit for organizations and creates an association with altruism in the customers' minds. Furthermore, Ar (2012) indicates that if the business focuses on innovating products and product environmental repercussions, it will gain an advantage over its rivals.

Authors Y. Chen et al. (2006) found that GPI is wholly linked with the organization's competitive advantage. GPI depicts the firm's vision, mission, and the green mindfulness of employees at every management level to stakeholders. Author Dangelico's (2016) findings highlighted the benefits as constituting a competitive advantage that improves market benefits, green reputation, and opportunities for innovation, leading to higher profits for the organization. The findings also show that the commitment of top management influences the development of green product innovation.

Furthermore, research on the influence of stakeholders found that foreign customers are significantly pressured to adopt a green strategy in product innovation (Ferrón-Vílchez 2016) due to international standards. Similarly, the authors Li et al. (2017) and Weng et al. (2015) stated that external legitimacy pressures such as ISO 14001 certification (Duque-Grisales et al. 2020; Ferrón-Vílchez 2016; Salim et al. 2018; Toha et al. 2020), competitors, and government pressure enhance the eco-friendly product practices among service and manufacturing companies (Chen 2020).

Alternatively, (Guoyou et al. 2013) found contrary evidence, suggesting that the community and institutional regulatory stakeholders have no significant effect on green product innovation. Authors Chang and Zhang (2019) also supported this with their research on green motives that influence green product innovation, showing that there is an insignificant influence on the product. They also found no decisive nexus in green co-production, green value in use, and green product innovation performance. The hypotheses are as follows:

**Hypothesis 1a (H1a).** *Green product innovation positively correlates with business performance (ROI, ROE, and ROA).*

**Hypothesis 1b (H1b).** *Green product innovation has a positive effect on sustainable development goal performance.*

2.2.2. Green Process Innovation (GPI)

The debate on the growing phenomenon of process innovation in corporations is due to the emergence of inductive thinking and creative design thinking, including new eco-friendly technology, green human resources, and green work practices to construct the structured approach to innovating the green production process. This can help the organization to maximize operational capability and focus on customer value, while also mitigating the environmental risk of production and operations.

Green process innovation is purposefully directed for the production process. Although it is new to the focal firm, it can diminish environmental risk and other repugnant consequences. The literature has recognized methods such as clean production (Ma et al. 2017), pollution control (Xie et al. 2019), pollution prevention (Wong et al. 2020), eco-competence, and circular technology. Green process innovation is initiated by adopting clean technology and eco-saving equipment to enhance energy efficiency, maximize resource utilization, and eliminate greenhouse gas emissions (Azevedo et al. 2017; Dai and Zhang 2017).

GPI is the second critical component of green innovation; it focuses on mitigating harmful environmental impacts through waste control, water management, and sustainable raw material procurement. (Khan et al. 2021; Khan and Johl 2019). Additionally, it improves organizational performance and reduces the organization's operating costs (Liu et al. 2020), allowing for revenue generation (Karabulut and Hatipoğlu 2020) and the development of trust among internal stakeholders (Khan and Johl 2020). This is because GPI eliminates incidents within the firm, thus, providing a secure workplace for its employees. It also benefits companies in terms of revenue and draws external stakeholders' attention to their firm's performance.

Ma et al. (2017) researched green process innovation, the effect on the firm's image, and its benefits (Zehir and Ozgul 2020). The researcher found a positive relationship with long-term benefits and a non-significant relationship in the short term (Khan et al. 2021). Authors Li et al. (2017), found that legitimacy pressure had a positive influence on green innovation. Despite the various benefits of green process innovation, numerous researchers claim that many firms lag in adopting green process innovation due to the lack of complete customer awareness, enforcement of green innovation by the government, and sustainable investment promotion (Dai and Zhang 2017).

On the contrary, there is an upward trend in the stock of most innovative or sustainable firms. For instance, Schneider Electric and Ørsted were recognized as top sustainable companies in 2021, increasing their stock price. Therefore, the following hypothesis is proposed:

**Hypothesis 2a (H2a).** *Green process innovation positively correlates with business performance (ROI, ROE, and ROA).*

**Hypothesis 2b (H2b).** *Green process innovation has a positive effect on sustainable development goal performance.*

2.2.3. Green Service Innovation (GSI)

Green service innovation is the third component of green innovation. It has drawn the attention of academia and industries due to the demand for competitiveness and under-researched variables of green innovation (Chang 2018). GSI (after-sales service) is less scrutinized by environmental regulators (Khan and Johl 2019). Furthermore, green service innovation can also be the source of minimizing the firm's cost of capital; green service innovation and incorporating the environmental perspective can reduce costs (Zhang et al. 2020) by making the firm stand out among its competitors. Due to the growing ecological awareness of stakeholders, companies must focus on other elements of green innovation and pay attention to GSI, which is also concerned with curbing environmental challenges and winning stakeholder confidence (Chang 2018) (Khan et al. 2021a).

The current model of service offers services to the customer through a third-party company. Manufacturing companies outsource service to other companies in order to provide services with minimum cost to the customer. Companies also believe outsourcing services will lead to expert service for their customers. Several kinds of research have been conducted on third-party services or servitization. Findings suggest that outsourcing services to third-party companies only minimizes the cost of the service provided but the sustainability by the third-party services is ignored (Cirpin and Kabadayi 2015). Offering services through a third party has been innovative; many researchers (Chu et al. 2018) and industrialists have accepted this yet remain doubtful about whether service providers are providing green services to customers.

The green service innovation debate influences the service industry and manufacturing industry. Most companies have realized the importance of green service innovation to achieve economic, social, and sustainable development, just as they have with every element of green innovation.

Authors Khan and Johl (2019, 2020) conducted a study on green service innovation, claiming that green service innovation is an essential element in achieving sustainability

that most studies have missed. Therefore, this research has adopted green service innovation in operational activities to investigate its effects on the performance of the companies (financial and non-financial). Thus, this study hypothesizes:

**Hypothesis 3a (H3a).** *Green service innovation positively affects the performance of the business (ROI, ROE, and ROA).*

**Hypothesis 3b (H3b).** *Green service innovation has a positive effect on sustainable development goals.*

*2.3. Green Non-Operational Innovation (GNOI), Firm SDGs, and Financial Performance*

The second significant variable is green non-operational innovation, which in this research includes green organizational innovation and green marketing innovation. Green non-operational innovation indirectly enhances revenue by minimizing the cost of capital through maximizing resource utilization such as sustainable electricity consumption, emission management, material utilization, and water management in non-production and manufacturing firms. In addition, green organizational innovation can be another practical element in generating revenue for the business and attracting responsible investment, boosting related stakeholders' confidence and creating a competitive advantage.

In addition, there is existing literature on non-operational innovation and financial performance (ROA, ROE) (Khan and Johl 2019, 2020); however, this is an early study investigating the relationship between green non-operational innovation and firm financial performance. In addition, this study is also among the first studies to examine the relationship between green non-operational innovation and firm sustainable development goal performance.

2.3.1. Green Organizational Innovation (GOI)

Green organization innovation is an innovation that supports operational innovation. This supporting innovation includes the procurement of raw materials, human resources, the development of infrastructure and technology, and collaboration. Green organizational innovation, or supportive innovation, supports innovation related to company products, processes, and services. Green organization innovation is gaining traction after various countries have made it mandatory for companies to report environmental, social, and governance (ESG) data.

These ESG data reflect the company's internal non-financial performance; these types of non-financial performance attract investors to the companies. Research conducted by the authors Brogi and Lagasio (2019) on the disclosure of environmental and social information in company reports has provided a significant and positive nexus between ESG and bank-related profitability.

The ESG disclosure of companies reports on the companies' products, processes, adoption of technology, infrastructure, culture, and innovation, since disclosure can affect the firm's profitability. Likewise, organizational innovation, which this research defines as support innovation, will affect the innovation of products, processes, and services, which will lead to profitability and increased firm performance. This research examines the raw materials, suppliers, infrastructure, technology used, organizational culture, and innovation strategies to measure green organizational innovation.

Furthermore, a variety of research has been conducted on innovation strategy (Brogi and Lagasio 2019; Cassiman and Veugelers 2006; Pisano 2015) and it has been found that the integration of innovation strategy with business strategy is the key to growth and success for an organization. Likewise, green innovation strategy research was conducted by (Song and Yu 2018), which resulted in a positive correlation between organizational identity and green creativity.

This green innovation strategy impacts the supportive activities of organization innovation such as adopting technology, building green infrastructure, the procurement of

green raw material, the development of green supply chain management, and emission minimization. An organization's innovation support activities are created through its green organization identity and green creativity culture. Research conducted by (Song and Yu 2018) on the integration of green innovation strategy, green organization identity, and green creativity found that green innovation strategy has a positive correlation in creating organizational identity in the form of a green creative culture. Authors Song and Yu (2018) also found that green organizational identity and green creative culture are positively related.

Therefore, to minimize emissions, waste, wastewater, and energy consumption, organizational identity and green creative culture play a vital role in creating a support system for the innovation of an organization's products, processes, and services. Alternatively, research conducted by (Xiao and North 2018) found that the supporting activity can only help in innovation, leading to an improvement in the products, processes, and services; however, this will not always lead to firm revenue generation. Therefore, the research hypotheses are as follows:

**Hypothesis 4a (H4a).** *Green organizational innovation positively correlates with firm performance (ROI, ROE, and ROA).*

**Hypothesis 4b (H4b).** *Green organizational innovation has a positive correlation with SDG performance.*

2.3.2. Green Marketing Innovation (G.M.I.)

In the last few decades, environmental awareness among consumers (Chen and Fang 2019) and the commitment to minimizing the effects from the operating activities of companies have led to green marketing innovation in order to gain benefits in terms of operations, market reputation, and customer attention (Chen and Fang 2019). The environment and innovation have emerged and are receiving growing attention in the marketing literature; the combination of marketing, environmental, and innovation concerns (Medrano et al. 2020) has given rise to a new approach in marketing products, processes, and services to the related stakeholders.

Although the green marketing area of focus in the literature is on packaging (D'Attoma and Ieva 2020), the analyses of consumers' purchasing (Fiore et al. 2017) have been limited to the product costs, quality, delivery, and green goodwill (Chen and Fang 2019) of the firm and its sustainable activities. This is due to the upsurge in stakeholder alertness of environmental obstacles in building the green goodwill of the firm by fulfilling the environmental requirements stated by (Khan and Johl 2019, 2020).

Green marketing innovation has been less of a focus of research in the green innovation literature. Green marketing innovation has been limited to green packaging (Ayandibu and Akbar 2021) or reusable packaging (Alabo and Anyasor 2020) and environmental certification (Chen and Fang 2019). Still, green marketing innovation can be thought of beyond packaging, ecological certificates, and calling back non-degradable plastic and glass containers that can be recycled by the same company.

In this context, quality certification and product certification are the best ways for the firm to fulfill the mandatory requirements of the environmental standard in the context of the market, but this will limit the firm's sustainable marketing to recycling. Reusable packaging should not be enough; a firm whose sustainable activities are limited to packaging is at the lower side of green marketing. Those who step beyond the recycling or reusing of packaging can be called mid-level and high-level green marketing firms.

This has resulted in addressing an increasing quest for developing innovative approaches to green marketing. An innovative approach to green marketing will make the firm stand out, which will create green goodwill for the firm. Green goodwill for the firm influences the customer and investor; however, it is not necessary to indulge them in buying and investment activities.

Therefore, this study will explore green marketing initiatives and approaches outside of recycled and reusable packaging.

**Hypothesis 5a (H5a).** *Green marketing innovation positively correlates with firm performance (ROI, ROE, and ROA).*

**Hypothesis 5b (H5b).** *Green marketing innovation has a positive relationship with sustainable development goals.*

*2.4. Conceptual Framework*

The above critical review provides a conceptual framework for establishing a background for the future research of proactive strategies to achieve maximum return, create a competitive advantage, and contribute to sustainable development goals. The independent variables of this research model are green innovation and reporting, which includes sub-dimensions such as green operational innovation and green non-operational innovation. Green operational innovation has three major variables: green product innovation, green process innovation, and green service innovation. Green non-operational innovation includes green marketing innovation and green organizational innovation. Furthermore, its dependent variables are the financial performance of companies and their performance in achieving sustainable development goals.

In the Figure 2, the independent variable of the research model includes all major variables of green innovation that exist in the literature due to the debate over green innovation classification in green operational and green non-operational innovations initiated by the authors Khan and Johl (2019). Additionally, this study extended the classification by incorporating green marketing innovation. Green marketing innovation (GMI), the missing variable in the classification, is imperative since GMI communicates the green initiative to stakeholders through advertising and green taglines. It also attracts new customers and puts pressure on rival firms in the industry.

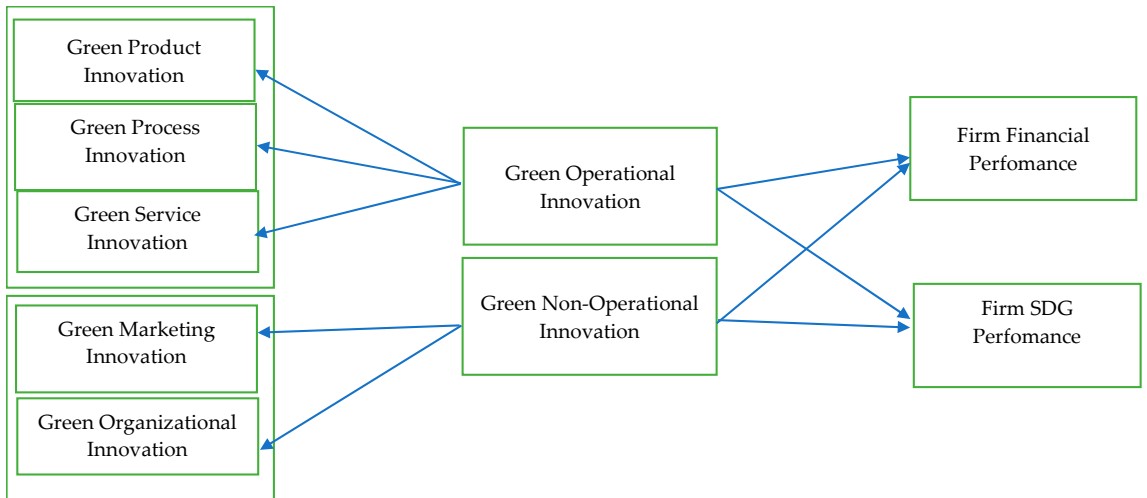

**Figure 2.** Research Model: Green Innovation Practices and Reporting Model.

Therefore, this study included all imperative variables of green innovation to test its relationship with the firm's financial performance and the firm's new, non-financial sustainable development goals. The next section of this research is the methodology adopted to test the formulated hypothesis and models developed in the above section.

**3. Materials and Methods**

To test the above hypothesis, this research is based on the Global Sustainable Responsible Investment report (GSRI) 2018. In the GSRI report (2018), the asset value was USD

22.90 billion in 2016, which rose to USD 30.7 billion in early 2018. In the same report, Europe and the United States were the highest out of Canada, Europe, Japan, Australia, New Zealand, and the United Nations. The lowest sustainable and responsible investment was by Australia, New Zealand, Canada, and Japan. However, the highest growth in sustainable and responsible asset management growth was seen in Japan. The detailed information is given in Table A1 Listed in Appendix A.

This study is in its primary stage, and from the five regions, this study will look at the top five countries and continents (based on G.D.P.) and blue-chip companies (based on market capital). Therefore, the total sample was 125; however, 75 firm reports were not in the English language. In total, the data from 67 firms were collected for final data analysis from every company's website for the years between 2018 and 2019. The summary of the 134 observations is shown in Table 1.

**Table 1.** Number of Observations.

| Continent | Country N | Company (N) | Number of Observations |
|---|---|---|---|
| Europe | 4 | 19 | 38 |
| Australia and New Zealand | 2 | 10 | 20 |
| Asia | 4 | 15 | 30 |
| North America | 3 | 11 | 22 |
| Africa | 3 | 12 | 24 |
| Total | | | **134** |

### 3.1. Content Analysis Procedure

Content analysis is widely used in assessing the business reporting literature. The main idea underlying this method is to search for corporate information and code it (numerically). This research has considered zero as not practice, whereas one is regarded as the company's practice (Xie et al. 2019). This research assumed that the companies disclose their green operational innovations and non-operational innovations in their sustainable and annual reports. The authors have downloaded the annual reports and sustainability reports of the companies and collected their website information (Zhang et al. 2020).

Therefore, the content analysis of the companies' reports was used to determine green innovation in different countries and regions. The content analysis for the level of green innovation in the organization is measured through the practices of green product innovation, green process innovation, green service innovation, green organization innovation, and green marketing innovation, which are given in Table A2.

### 3.2. Instrument Development

Instrument development and validation involve three phases: (1) the construction of the conceptual framework and item generation; (2) judgment quantifications; (3) psychometric testing of instrument properties (instrument reliability and validity) suggested by the authors Kääriäinen et al. (2020). After extensively reviewing the literature measuring green innovation, the measurement item was generated and ranked as per repetition of the item in the exiting instrument. The highest recurring item was adopted for a green product, green process, and green organizational innovation, whereas green services and green marketing were adopted from the research (Khan et al. 2021).

The first dependent variable, financial performance, is measured through financial ratios, namely, return on investment, return on equity, and return of investment (Xu et al. 2021; Yi et al. 2021), to evaluate the firm's performance. The other dependent variable, the firm's sustainable development goals, is adopted from the literature (Khan et al. 2021) to measure the firm's contribution to the United Nations' 17 SGDs.

The second phase of instrument development was judgment quantification by academic, industry, and policymaking experts, and suggestions were incorporated and improved to capture the firm's green innovations and reporting highlighted in Table A2: Detailed Criteria in Selection Variable Measurement.

In the third phase, psychometric testing of instrument properties (instrument reliability and validity) was conducted, using the reliability test of green innovation with a Cronbach's alpha of 0.755, which is more than the rule of thumb for good reliability statistics above 0.7. A similar approach was also adopted in the research (Chen 2016). After collecting and entering the data, this research explored, examined, and addressed outliers using the SPSS explore command to identify outliers and influential observations in the dataset; in total, 24 observations were eliminated from the final observation for data analysis.

### 3.3. Model Development

Panel data were used in the current study for analyzing the impact of green innovations on a firm's financial performance and SDGs. A panel regression is different from a general time series and cross-sectional regression model as it incorporates a double subscript. Another significant advantage of using panel regression is that it helps control unobserved variables, which change over time but not among entities. In addition, in panel model estimation, the time effect is also included, which helps in controlling individual heterogeneity by allowing firm-specific random or fixed effect components (Baltagi 2008).

There are many reasons for employing panel data estimation in the current study. Firstly, in the panel model, firms are contemplated as heterogeneous, whereas this is not the case in time and cross-sectional data series, which result in biases. Therefore, the main reason is the ability in managing heterogeneity. Secondly, the panel data approach provides more variation in datasets, high information data, and less multicollinearity with high efficiency and degree of freedom (Gujarati and Porter 2009). The model used in this research consists of n cross-sectional units, $n = 1, \ldots, N$ observed at each $t$ period, $t = 1, \ldots, T$. The total observation in the dataset is $n \times t$. Past research has constructed a panel data structure (Akhtar et al. 2020). The following panel regression model uses the same panel dataset structure as designed by the mentioned researchers (Akhtar et al. 2020):

$$y_{nt} = \alpha + \beta x_{nt} + e_{nt} \tag{1}$$

where $y_{nt}$ refers to regress, $\alpha$ refers to the intercept term, $\beta$ is K × 1 vector of the parameter to be estimated, and $x_{nt}$ is the $n$th observations on K regressors, which is 1 × k, $t = 1, \ldots, T$, $n = 1, \ldots, N$. The operational form of the model is:

Financial performance/SDGs = f (green operational innovations, green non-operational innovations)　　(2)

where ROA, ROE, and ROI measure financial performance and the 12 SDGs formulated by the United Nations. Green product, green process, and green service innovations are the parameters of green operational innovations. Green marketing and green organizational innovations are the parameters of non-operational innovations. The following four models were developed to analyze the impact of green innovations on firm financial performance and SDGs by boarding the proxies used in Equation (2):

Model 1: $ROA_{nt} = \alpha_n + \beta_1 GPI_{nt} + \beta_2 GPRI_{nt} + \beta_3 GSI_{nt} + \beta_4 GMI_{nt} + \beta_5 GOI_{nt} + \varepsilon_{nt}$

Model 2: $ROE_{nt} = \alpha_n + \beta_1 GPI_{nt} + \beta_2 GPRI_{nt} + \beta_3 GSI_{nt} + \beta_4 GMI_{nt} + \beta_5 GOI_{nt} + \varepsilon_{nt}$

Model 3: $ROI_{nt} = \alpha_n + \beta_1 GPI_{nt} + \beta_2 GPRI_{nt} + \beta_3 GSI_{nt} + \beta_4 GMI_{nt} + \beta_5 GOI_{nt} + \varepsilon_{nt}$

Model 4: $SDG_{nt} = \alpha_n + \beta_1 GPI_{nt} + \beta_2 GPRI_{nt} + \beta_3 GSI_{nt} + \beta_4 GMI_{nt} + \beta_5 GOI_{nt} + \varepsilon_{nt}$

where:
   $n$—Represents individual,
   $t$—Indicates years,

$\varepsilon_{nt}$—Random error term,

$\alpha_n$—Constant term,

$\beta_n$—Co-efficient of independent variables.

Moreover, the details of all other variables are shown in Table A2.

In addition, the current study has employed the fixed and random effects model for estimating panel data equations. Furthermore, Hausman's test is employed to decide between random or fixed effects; the null hypothesis is random effects and is an appropriate model.

## 4. Result

Table 2 presents the descriptive statistics of the major variables adopted in the research such as green operational and non-operational innovation (predictive variables), namely, green product innovation, green process innovation, green service innovation, green market innovation, green organizational innovation, and the dependent variables ROE, ROI, ROA, and SDGs.

**Table 2.** Descriptive Statistics.

| Variable | Obs. | Mean | Std.Dev. |
|---|---|---|---|
| ROE | 134 | 39.233 | 222.858 |
| ROI | 134 | 13.161 | 13.511 |
| ROA | 134 | 6.472 | 7.06 |
| SDGs | 134 | 0.338 | 0.354 |
| GPI | 134 | 0.321 | 0.236 |
| GPRI | 134 | 0.576 | 0.221 |
| GSI | 134 | 0.142 | 0.242 |
| GMI | 134 | 0.015 | 0.069 |
| GOI | 134 | 0.416 | 0.218 |

Source: Author's Calculation (STATA).

Table 3 presents the Pearson correlation coefficients of the variables. The result highlights that all the green operational innovation variables are significantly correlated with SDGs, and none of the green non-operational innovation variables are significantly correlated with SDGs. However, when it comes to firm performance, GPI, GSI, and GMI are significantly correlated with ROE; GPI, GPRI, and GOI are significantly correlated with ROI, whereas GPRI, GSI, and GMI are considerably correlated with ROA.

**Table 3.** Pearson Correlation Matrix.

| VARIABLES | (1) | (2) | (3) | (4) | (5) | (6) | (7) | (8) | (9) |
|---|---|---|---|---|---|---|---|---|---|
| **(1) ROE** | 1.000 | | | | | | | | |
| **(2) ROI** | 0.624 *** | 1.000 | | | | | | | |
| **(3) ROA** | 0.072 | 0.531 *** | 1.000 | | | | | | |
| **(4) SDGs** | −0.099 | −0.109 | 0.033 | 1.000 | | | | | |
| **(5) GPI** | 0.206 ** | 0.165 * | −0.031 | 0.239 *** | 1.000 | | | | |
| **(6) GPRI** | 0.151 | 0.057 ** | −0.159 * | 0.287 *** | 0.677 *** | 1.000 | | | |
| **(7) GSI** | 0.145 * | −0.012 | −0.115 ** | 0.196 ** | 0.527 *** | 0.440 *** | 1.000 | | |
| **(8) GMI** | −0.009 * | −0.030 | 0.021 * | 0.065 | 0.135 | 0.064 | 0.172 ** | 1.000 | |
| **(9) GOI** | 0.135 | 0.152 * | −0.001 | 0.129 | 0.625 *** | 0.491 *** | 0.354 *** | 0.566 | 1.000 |

Correlation is significant at 1% (***), 5% (**), and 10% (*). Source: Author's Calculation (STATA).

In static panel regression, the major issue is the presence of multicollinearity. Our dataset comprises 67 cross-sections with two years, so it becomes essential to test the variance inflation factor (VIF) before running the regression. It was found that the data is free from multicollinearity because the result depicts all the variables as having a value less than ten (which is reflected in Appendix A Table A4).

Next is the heteroscedasticity test to check the variances in error terms. The Breusch–Pagan test is conducted for heteroscedasticity (Gujarati and Porter 2009). Table A3 below shows that the first three models have heteroscedasticity, and the last model has homoscedasticity. The first three models' alternate hypotheses are accepted, whereas in the last model, null is accepted (null: homoscedastic and alt: heteroscedastic). This study has used the Huber–White Sandwich's robust technique in regression analysis to deal with heterogeneity problems (Huber 1967).

Before explaining the results, it is necessary to clarify that after running fixed effect and random effect regression, the Hausman test was conducted. The Hausman results showed the selection of random effect regression in all the models as the value was more than 0.05 (null: random; alt: fixed). In addition, because of heteroscedasticity in the first three models, a random robust regression technique was employed to interpret the results. Moreover, for the fourth model, the normal random effect model was taken into consideration as it was free from heteroscedasticity.

Table 4 illustrates the regression result of the (causal relation) of green operational elements, namely, green product innovation (GPI), green process innovation (GPRI), green service innovation (GSI), and green non-operational variables, namely, green marketing innovation (GMI) and green organizational innovation (GOI). It depicts the four crucial models of this study, explaining the mixed positively and negatively correlated results.

**Table 4.** Regression Results.

| Variables | Financial Indicators (Robust Techniques) | | | SDG (Sustainability) |
| --- | --- | --- | --- | --- |
| | Model 1 (ROA) | Model 2 (ROE) | Model 3 (ROI) | Model 4 (SDGs) |
| GPI | 1.25 (4.503) | 0.24 ** (3.617) | 1.77 * (13.928) | 0.13 ** (1.923) |
| GPRI | −2.36 ** (−7.786) | 0.31 (4.043) | 0.26 *** (3.912) | 1.97 * (0.367) |
| GSI | −1.76 * (−3.358) | 0.17 * (2.870) | −1.19 (−6.917) | 0.68 (0.100) |
| GMI | 0.59 ** (3.391) | −0.77 ** (−3.943) | −0.60 (−7.774) | 1.88 * (0.358) |
| GOI | 0.57 (2.059) | 0.25 (5.686) | 1.03 (5.791) | −0.56 (−0.098) |
| Constant | 3.98 *** (9.082) | −0.65 (−9.260) | 3.12 *** (10.784) | 1.21 (0.112) |
| R-Squared | 0.128 | 0.146 | 0.152 | 0.193 |
| F-Test | 2.119 * | 4.412 *** | 4.521 *** | 2.618 |
| N | 134 | 134 | 134 | 134 |

Significant at 1% (***), 5% (**), and 10% (*). Source: Author's Calculation (STATA).

Model 1 has a positive relation to green marketing innovation; however, it is negatively significant with regard to green process innovation, green service innovation, and the return on assets (ROA). In contrast, green product innovation and green process innovation significantly impact sustainable development goal performance. Furthermore, Model 1 does support the existing literature on green product innovation, which is found insignificant in the short term. The green process innovation and green service innovation are still negatively significant because the shift in processes and services requires enormous investment. However, the return is reflected in the firm balance sheet in the long term (Ma et al. 2017), as reported in the literature.

Therefore, Model 1 does not support green innovation with ROA but significantly supports ROE and ROI (Hypothesis 1a) and supports hypothesis H1b. Green product innovation is insignificant to the ROA but highly significant to the firm's sustainable development goals in the short term.

The better result can be seen in Model 2 for the green product innovation along with green service innovation (GSI), which is positively significant for the return on equity (ROE) (Xie et al. 2019). In contrast, green marketing innovation (non-operational) is negatively related, and green process innovation is insignificant. The logic behind green products and green services is they can be marketed well directly, attracting responsible investment. Still, for marketing expenditure spent in the year, the return is expected in the coming years as

advertising expenditure is treated as capital expenditure, affecting profit and dividends and, thus, returns on equity.

In Model 3, green product innovation and green process innovation are significantly positive for the return on investment (ROI) as they immediately improve the product and production process (Wang et al. 2021). In contrast, green services, green marketing, and green organizational behavior remain insignificant to the investment. The fourth model (Model 4) also has a similar result for green product innovation and green process innovation, which are significant for sustainable development goals and green marketing innovation in the short term (i.e., two years). In contrast, the other operational variables, green service innovation and green organizational innovation, remain insignificant.

### 5. Green Innovation Disclosure (Reporting)

The reporting of green innovation (green operational innovation and non-operational innovation) has been seen mainly in Europe, followed by North America. The least reporting of green innovation has been observed in Africa, followed by Australia. The authors explored the detailed disclosure of green innovation variables in all five continents, which is deployed in the pie diagram given below.

In regard to the first variable, green product innovation disclosure has been discussed as being 28% in Australian and New Zealand companies, followed by European companies at 27%. The lowest green product innovation disclosure is in Africa at 8%, Asia at 17%, and North America at 20% in blue chip firms. The other variable of green innovation is green operational innovation, of which Australia and New Zealand, at 28%, are leaders in disclosing green process innovation, followed by Europe at 21%. North America and Asia remain the same at 17% disclosure. In contrast, Africa remains the lowest out of the 67 companies from each of the top five countries and continents.

The last variable of green operational innovation is green service innovation (Figure 3c), of which the highest disclosure is now in Europe, a leader at 41%, followed by North America at 39%, and Australia and New Zealand at 14%. Nevertheless, Africa remains in the lowest position at 6%, followed by Asia at 0% in blue chip firms.

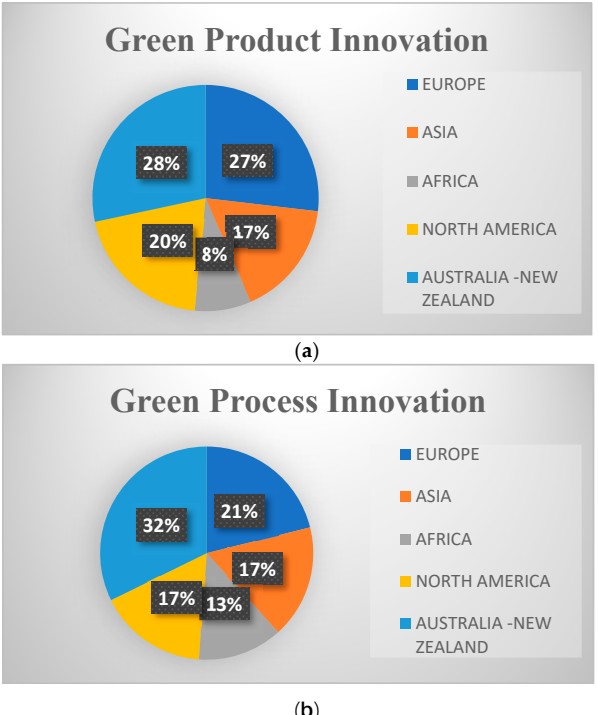

(**a**)

(**b**)

**Figure 3.** *Cont.*

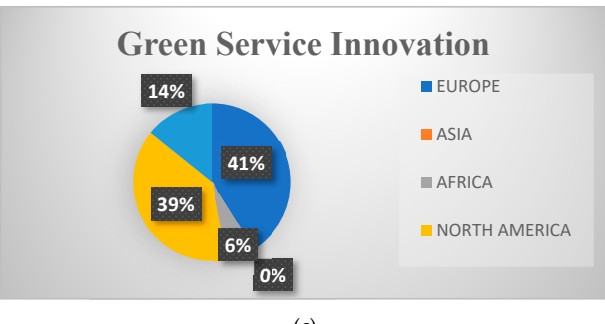

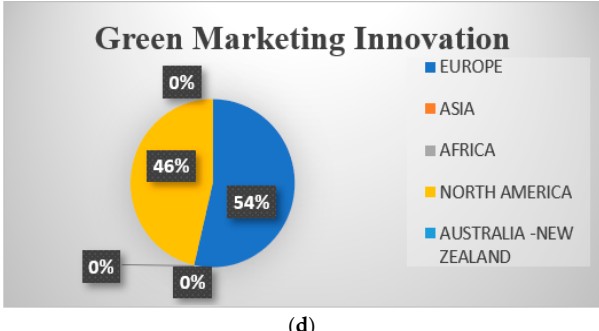

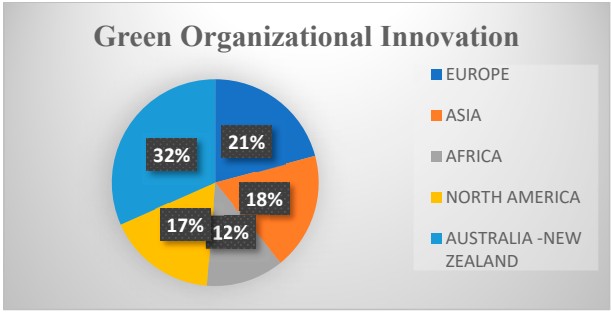

(e)

**Figure 3.** (**a**) Green product innovation reporting; (**b**) green process innovation reporting; (**c**) green service innovation reporting; (**d**) green marketing innovation reporting; (**e**) green organizational innovation reporting.

The second variable in this research of green innovation is green non-operational innovation. Two major variables are green marketing innovation and green organizational innovation; these variables are treated as a support system for green operational innovation. In Figure 3d, green marking innovation disclosure is led by Europe at 54% and North America at 46%. In contrast, for the other three regions of North America, Africa, and Asia, disclosure remains at zero (0%) out of 67 companies from the top five continents and countries.

The last variable of green non-operational innovation is green organizational innovation; in Figure 3e, the highest percentage of disclosure is Australia and New Zealand at 32%, Europe at 21%, and Asia and North America remain the same at 17%–18%. In contrast, Africa remains the lowest out of the 67 companies from each of the top five continents and countries. Therefore, the leader in disclosing green innovation practices is Australia and New Zealand, followed by Europe in second place, then North America in third place, Asia in fourth, and, finally, Africa.

## 6. Discussion and Conclusions

The increase in environmental and social challenges has raised concern among researchers and policymakers. To mitigate these challenges, the business approach has been investigated in the existing literature with significant results; however, there is still a rise in emissions, waste, water contamination, and so forth. The increase in emissions, waste, and water contamination have provoked firms to shift from reactive to proactive approaches. This research has advocated and strategically tested green innovation practices positively with firm financial performance and sustainable development goals.

Moreover, this research expressed the imperativeness of a proactive approach within the business' operational and non-operational innovation and creative collective approach to contributing to the United Nations' SDGs at a firm-level without compromising firm financial performance. The adoption of SDGs as non-financial parameter measurement criteria will create a collective approach in meeting the 17 SDGs.

On the contrary, reporting is always considered a way of communicating with their related and unrelated stakeholders. The literature on non-financial reporting has mostly focused on the operational Carbon Discloser Project (CDP) (Lombardi et al. 2021; Wójcik-Jurkiewicz and Sadowska 2018), and environmental disclosure (Aluchna and Roszkowska-Menkes 2019) of firm performance. The current approach of reporting used by businesses in their sustainable activities tends to be a normative approach and less transparent in terms of firm innovation activities. However, there has been a growing demand by stakeholders to incorporate a proactive approach in business activities. To meet that, green innovation practices (GIPs) have been integrated as a new element in environmental reporting, thereby reflecting the proactiveness of companies in adopting green innovation reporting (G.I.R.).

This study aims to advocate for greater disclosure on sustainability reporting by incorporating the GIPs of companies and substantial, sustainable development goal contribution. The adoption of green innovation reporting will improve the firm's sustainable creation of target results while also increasing the accountability of business operations, stakeholder trust, and the sustainable development of goal performance. This part of the research is further divided into two important discissions and implications, i.e., the practical implications and the implications for policymakers.

### 6.1. Practical Implication

This research contributes to the most widely used pollution-associated industries such as chemical, oil and gas, agriculture, and textiles, which are responsible for pollution on a large scale. For instance, greener solvents such as ionic liquids are being used in the in the production of pharmaceuticals (Khan et al. 2021b). Also, the use of green emulsion liquid membrane for the removal biologically active drug molecules from the wastewater (Khan et al. 2020a); (Khan et al. 2021a) and production of biodiesel from waste vegetable oil to mitigate the emission (Khan and Athar 2015). Another imperative example is in the oil and gas industry, where the various incidents of oil spills have become disastrous for aquatic life. Green ionic liquid can be used to deal with cleaning oil spills (Khan et al. 2020b; Nazar et al. 2021). Lastly, a mixed matrix membrane is an emerging technology that purifies biogas (Abd Rahman et al. 2018). This shows the practical application of green innovation in operational and non-operation activities that mitigate more significant environmental challenges.

According to the findings, implementing environmental business practices such as green innovation and reporting will help businesses gain a competitive edge and improve their organizational and ecological efficiency. For instance, the operational applicability of green innovation, disclosure, and the sustainable development goals model can be seen in various sectors, including wastewater from pharmaceuticals, oil spills, and dye wastewater from the textile industry.

In dealing with oil spills, businesses often do not announce the creative measures taken to clean up and minimize oil spills in their annual reports. Oil and gas companies disclosing green innovation steps in dealing with environmental issues, the invested amount, and

future measures to avoid such incidents in their reports will boost the firm's goodwill. This increases the investors' confidence and convinces them to believe that the firm will not cause oil spills and incur environmental fines in the future. Consequently, its triple bottom line would be unaffected, allowing it to retain long-term, responsible investors. The attraction of sustainable, responsible investors will directly contribute to the firm's sustainable development goals and stable financial performance.

*6.2. Policy Making Implications*

Environmental issues and data transparency, such as greenhouse gas emissions, waste control, water management, and other topics, are subjects addressed by the United Nations as they aim to reduce environmental pollution and its effects on society. Due to ecological challenges, different strategies have been developed and adopted, such as integrating reporting, the publication of which has been made mandatory for all companies in various countries, and various reporting standards (global reporting standards), which are formed from volunteering the transparency of business practices to the stakeholders.

This research draws the attention of policymakers and the global reporting initiatives towards including green innovation disclosure as part of the standards for integrating reporting since green innovation activities can minimize environmental challenges at the innovative stage and mitigate existing environmental challenges.

Secondly, policymakers can also see the importance of adopting GIR in achieving the 17 imperative sustainable development goals of the United Nations in this research, which has shown positive findings of GIR on the sustainable development goals of firms. Mainly, it includes climate action (SDG 13), responsible consumption and production (SDG 12), industry, innovation, and infrastructure (SDG 9), and decent work and economic growth (SDG 8) as stated by the authors M. Zhou et al. (2020).

Furthermore, this research will help policymakers see how the operational activities of businesses indirectly contribute to the other SDGs such as good health and well-being (SDG3), clean water and sanitation (SDG6), affordable clean energy (SDG7), sustainable cities and communities (SDG11), life below water (SDG14), life on land (SDG15), and promoting peaceful and inclusive societies (SDG16).

Lastly, this research draws significant attention to the management team of the United Nations' 17 sustainable development goals, which is using green innovation practices and reporting to embark upon the SDGs at the firm-level. These research findings have shown a direct contribution to sustainable development goals.

## 7. Limitation and Future Research

The study has adopted three accounting ratios to measure firm financial performance. Future research can also adopt Tobin's Q to address the market perspective. Additionally, more regress analyses (G.M.M., 2LS, and 3LS) can be adopted in future research for generalizing the results. In addition, green innovation (green operational and non-operational) activities can be further investigated in the moderating role of environmental managerial and directorial concerns.

**Author Contributions:** Conceptualization, P.A.K.; data curation, P.A.K.; formal analysis, P.A.K. and S.A.; investigation, P.A.K. and S.A.; methodology, P.A.K.; supervision, S.K.J.; validation, P.A.K.; writing—original draft, P.A.K. and S.A.; writing—review and editing, P.A.K., S.K.J., and S.A. All authors have read and agreed to the published version of the manuscript.

**Funding:** This study is funded by the matching fund of U.T.P. Malaysia and Uhamka Indonesia (Grant No.: 015ME0-174).

**Data Availability Statement:** Available upon request.

**Acknowledgments:** The authors would like to thank the M.H. Department of Universiti Teknologi PETRONAS, Malaysia, for facilitating the support to conduct this research.

**Conflicts of Interest:** The authors declare no conflict of interest.

## Appendix A

**Table A1.** Source: Global Sustainable Investment Review-2018.

| Region | 2016 | 2018 |
|---|---|---|
| Europe | 12,040 | 14,075 |
| United States | 8723 | 11,995 |
| Asia | 474 | 2180 |
| Canada | 1086 | 1699 |
| Australia/New Zealand | 516 | 734 |

**Table A2.** Detailed Criteria in Selection Variable Measurement.

| Variable | Measurement Criteria | Data Source | Reference |
|---|---|---|---|
| Independent Variables | | | |
| Green Operational Innovation | | | |
| Green Product Innovation (GPI) | Product Innovation-Green<br>Green Product Innovation Goal<br>ISO 14001 Certification<br>Disassembly and Disposal<br>Eco-Labeling<br>Lifecycle effect on the environment<br>Continues Improvement /Innovation<br>Green Packaging<br>Emission Intensity (Per product) improvement | Sustainability Reports | (Das et al. 2000) |
| Green Process Innovation (GPRI) | Emission of Waste, Efficiency of energy, green materials, green technology, Emission minimization, and Green Business Certification Inc (GBCI) | Sustainability Reports | (Palčič and Prester 2020) |
| Green Service Innovation (GSI) | Green Service Goal, Green technology adoption in service and Green Material used in services | Sustainability Reports | (Khan et al. 2021) |
| Green Non-Operational Innovation | | | |
| Green Organization Innovation (GOI) | Green Building certification, Green Business Certification Inc (GBCI), and SITES Certification | Sustainability Reports | G.R.I. |
| Green Marketing Innovation (G.M.I.) | Green Advertisement, Green Tag Line, No Greenwashing | Sustainability Reports | (Khan et al. 2021) |
| Dependent Variables | | | |
| Firm Financial Performance | Return on Assets (ROA), Return on Equity (R.O.E.), Return of Investment (R.O.I.). | Annual Report and Financial Report | (Xu et al. 2021; Yi et al. 2021) |
| Sustainable Development Goal | | Sustainability Reports | (Khan et al. 2021) |

**Table A3.** Heteroscedasticity Test.

| DEPENDENT VARIABLE | Chi2(1) | Prob. > Chi2 |
|---|---|---|
| MODEL 1—ROA | 16.98 | 0.0000 *** |
| MODEL 2—ROE | 290.70 | 0.0000 *** |
| MODEL 3—ROI | 69.28 | 0.0000 *** |
| MODEL 4—SDGs | 2.52 | 0.1126 |

**Note:** *** Statistically significant level at 1%. **Source:** Author's Calculation (STATA).

**Table A4.** VIF (Variance Inflation Factor).

| VARIABLES | VIF | 1/VIF |
|---|---|---|
| GPI | 2.590 | 0.387 |
| GPRI | 1.910 | 0.524 |
| GSI | 1.670 | 0.600 |
| GMI | 1.430 | 0.697 |
| GOI | 1.040 | 0.963 |
| **Mean** | **1.730** | |

**Source:** Author's Calculation (STATA).

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
