# Peer review of "Firm Sustainable Development Goals and Firm Financial Performance through the Lens of Green Innovation Practices and Reporting: A Proactive Approach"

_jrfm, doi:10.3390/jrfm14120605_

Round 1
Reviewer 1 Report
Dear Authors,
Thank you for the opportunity to read and review your manuscript submitted to the Journal of Risk and Financial Management. After reading the manuscript I can clearly see that you have accomplished huge research which undoubtedly gives relevant findings. However, there are some issues that stop me from being convinced that the quality of the current version is suitable for publication. Therefore, I advise major revision. Here is the list of the recommendations and comments:
- In the manuscript you mention ‚author‘, ‚scholar‘, but there is no specification of who was the author (e.g. line 58, 169, 188, etc.).
- It is unclear how you came to the formulation of hypotheses. These hypotheses lack a more specific scientific background. I mean how did you come to the hypotheses that every type of innovation is related to ROI, ROA, ROE when the literature review does not include texts about the possible relations with these measures. The same applies to the sustainable development goal performance.
- The classification of innovations lacks clarity. Why non-operational innovations consist of green marketing innovation and green organizational innovations. Isn't it an overlapping classification? I would consider green marketing innovation as a part of green organizational innovation. Green marketing innovations in some cases might be of operational level, so why this type of innovation belongs only to non-operational block. I am not trying to convince you to change the classification, but a strong scientific background for such kind of choices is essential. Therefore, I encourage you to strengthen the scientific reasoning of classification.
- Why have you mentioned ‚banks‘ in line 465?
You will be able to find more specific comments in the attachment.
Once again, thank you for the opportunity, and wish you good luck in strengthening the manuscript.

Author Response
Thank you so much for your critical comments, we can see the improvement in the manucript after incorporating the comments. We all authors thank full of you for this contribution.
Thank you so much

Reviewer 2 Report
It is suggested that the authors restructure the paper to achieve better flow and soundness (deductive reasoning, argumentation), and upgrade it with additional references (for example: Azevedo, S.G.; Godina, R.; Matias, J.C.d.O. Proposal of a Sustainable Circular Index for Manufacturing Companies. Resources 2017, 6, 63. https://doi.org/10.3390/resources6040063; ĐURIĆ, Zorica, POTOČNIK TOPLER, Jasna. The role of performance and environmental sustainability indicators in hotel competitiveness. Sustainability; etc.).
Author Response
Thank you so much for your valuable suggestion, we have rearranged the few section. we can the see the positive impact in doing.
Thank you so much

Reviewer 3 Report
Review comments for “ Firm Sustainable Development Goal and Firm Financial Perfor mance from the Lens of Green Innovation Practices and Reporting: A proactive approach”(jrfm-1437858)
This research aimed to investigate the effect of green innovation practices towards firm sustainable development 12 goals (S.D.G.) and firm financial performance simultaneously. This study also aimed to inject green innovation reporting into sustainable reporting for greater disclosure. Overall, the paper is well written and deals with an important topic. Yet, I found several problems that the authors should deal with. My comments are as follows:
- The literature review section is too lengthy. I also found some redundancies. The authors should make the manuscript more concise. Many paragraphs are descriptive. In-depth discussion is recommended in this section.
- This manuscript definitely needs a professional editing by a native speaker whose first language is English. I found many grammatical and editorial errors. In addition, the quality of writing throughout the manuscript is somewhat low.
- Regarding the conceptual model, why study variables are included and located in the hypothesized model is feebly justified. The proposed model looks more like the collection of existing important variables. This is the major weakness of this research.
- Is there any reason to locate two dependent variables as final constructs? The proposed theoretical framework lost the focus due to this reason.
- Some tables are not necessary.
- Discussion and conclusion are somewhat weak. The authors failed to demonstrate why this research was necessary and why this research is important.
Hope my comments described above help the authors improve the quality of the paper. I wish you luck for the manuscript revision.
Author Response
Dear Reviewer,
We authors are grateful to you for your valuable sussgtion, and we clearly see the improvements in the article. We are blessed to have such constructive comments.
Thank you so much

Round 2
Reviewer 1 Report
Dear Authors,
Thank you for the revised version of the paper. The improvements are significant, however, several minor recommendations should be regarded before publication. As I have already mentioned in the previous review, in the manuscript you mention ‚author ', ‚scholar ', but there is no specification of who was the author (e.g. line 58, 169, 188, etc.). Please specify the authors, scholars by indicating their surnames in the texts.
Author Response
|
Apologies, thank you so much for highlighting again, we have incorporated all the suggestion. Really, I am thankful of all your critical comments. |

Reviewer 2 Report
No additional references were added as it was recommended by one of the reviewers, and authors did not explain their decision not to follow the recommendation.
Author Response
|
Thank you so much, this round of review I have included recommended reference. In line number 35 and 222 |

Reviewer 3 Report
I appreciate that the author(s) addressed my comments logically and thoroughly. I believe that the revised manuscript is now suitable to be published in the JRFM.
Author Response
Thank you so much, I see the importance of your suggestion, the article has become more objective oriented, readable, and standard article.
Thank you so much from bottom of my heart
